

# Toxicity thresholds of three insecticides and two fungicides to larvae of the coral *Acropora tenuis*

Florita Flores[1], Sarit Kaserzon[2], Gabriele Elisei[2], Gerard Ricardo[1] and Andrew P. Negri[1]

[1] Australian Institute of Marine Science, Townsville, QLD, Australia
[2] Queensland Alliance for Environmental Health Sciences (QAEHS), The University of Queensland, Woolloongabba, QLD, Australia

## ABSTRACT

Tropical marine ecosystems, such as coral reefs, face several threats to their health and resilience, including poor water quality. Previous studies on the risks posed by pesticides have focused on five priority herbicides; however, as the number of pesticides applied in coastal agriculture increases, a suite of 'alternative' pesticides is being detected in tropical nearshore waters. To improve our understanding of the risks posed by alternative pesticides to tropical marine organisms, the effects of three insecticides (diazinon, fipronil, imidacloprid) and two fungicides (chlorothalonil, propiconazole) were tested on larval metamorphosis of the coral *Acropora tenuis*. *A. tenuis* larvae were affected by all five pesticides and the reference toxicant copper. The no effect concentration (NEC) and the 10% and 50% effect concentrations (EC$_{10}$ and EC$_{50}$, respectively) for larval metamorphosis were estimated from concentration-response curves after 48 h exposure. The NEC, EC$_{10}$ and EC$_{50}$ (in µg L$^{-1}$), respectively, of each pesticide were as follows: chlorothalonil (2.4, 2.8, 6.0); fipronil (12.3, 13.9, 29.1); diazinon (38.0, 40.8, 54.7); imidacloprid (263, 273, 347); and propiconazole (269, 330, 1008). These toxicity thresholds are higher than reported concentrations in monitoring programs; however, these data will contribute to improving water quality guideline values, which inform the total risk assessments posed by complex contaminant mixtures to which these pesticides contribute.

# INTRODUCTION

## Pesticides in tropical marine waters

Tropical marine ecosystems are under intense pressure from global climate change (*Hughes et al., 2018*), compounded by local pressures, including poor water quality from coastal development (*Heery et al., 2018*; *Waterhouse et al., 2012*). In Singapore, for example, the average water visibility has decreased from 10 m to 2 m due to sediment input from land reclamation and coastal development (*Heery et al., 2018*). Intensive coastal agriculture also poses a growing threat to tropical marine ecosystems worldwide, with models predicting that by 2050 about 1 billion ha of additional land would need to be

Corresponding author
Florita Flores, f.flores@aims.gov.au

converted into agriculture to meet global demands (*Laurance, Sayer & Cassman, 2014*). Tropical marine habitats off the coast of Nicaragua are severely degraded due to land clearing of coastal forest cover leading to increased sedimentation and degrading water quality (*Jameson et al., 2019*). Likewise, the coastal ecosystems of Malaysia are under continued threat by land reclamation and domestic and industrial waste pollution (*Sany et al., 2019*). In Australia more than 80% of the Great Barrier Reef (GBR) catchment area supports some form of agriculture (*Gilbert & Brodie, 2001*), which is dominated by sugarcane cultivation near the coast (*Lewis et al., 2009*; *Waterhouse et al., 2012*). Wet-season runoff from agricultural and urban development activities has reduced the water quality of the GBR over successive decades (*Furnas, 2003*; *Waterhouse et al., 2012*). Along with elevated sediment and nutrients, pesticides (herbicides, insecticides and fungicides) from agricultural industries represent ongoing hazards to ecosystems of the nearshore GBR lagoon (*O'Brien et al., 2016*; *RWQIP, 2018*). The GBR represents one of the most comprehensively monitored tropical marine systems with respect to pesticides, and it has been estimated that over 17,000 kg of the six most widely applied Photosystem II (PSII) herbicides (ametryn, atrazine, diuron, hexazinone, simazine and tebuthiuron) enter the GBR annually (*Brodie et al., 2017*). A recent study by *Warne, Smith & Turner (2020)* analysed over 2,600 water samples from 15 waterways that enter the GBR lagoon and found 99.8% of the samples had detectable concentrations of pesticides and pesticide mixtures.

## Alternative pesticides in the GBR

Successive Australian government programs have aimed to reduce the loads of pesticides entering waters of the GBR and its catchments (*Brodie et al., 2017*; *RWQIP, 2018*). At the same time, there have been changes in regulations and registrations for pesticide application in coastal agriculture (*Davis et al., 2014*), leading to shifts in usage patterns and to at least 44 'alternative' pesticides being detected in the GBR catchments and lagoon (*King, Alexander & Brodie, 2013*; *O'Brien et al., 2016*). For example, the insecticides chlorothalonil, fipronil and propiconazole have been detected in the catchments while imidacloprid and diazinon have been found in both the catchments and GBR lagoon (*Devlin et al., 2015*; *O'Brien et al., 2014*). More specifically, imidacloprid is the most frequently detected insecticide in the GBR lagoon, being present in over half of the samples (50.3%) that were analysed for this insecticide between 2011 and 2015 (*Warne, Smith & Turner, 2020*). Currently, fipronil is not included in the pesticide analytical suite for marine samples by the Great Barrier Reef Marine Park Authority Marine Monitoring Program (GBRMPA MMP) (*Gallen et al., 2019*); however, it is used in the catchments and should be included in future monitoring of marine samples. With the continued improvement of detection methods, it is likely that additional pesticides will be detected in tropical marine waters in the future (*Devlin et al., 2015*).

## Guidelines for alternative pesticides

Monitoring and reporting concentrations of pesticides in tropical marine waters represents an important contribution to effectively manage long-term reductions in pesticide

concentrations and loads (*Brodie et al., 2017*). However, to successfully assess the risks to tropical marine ecosystems we also need to understand the toxicity threshold for each pesticide to both the individual species (especially key habitat builders, such as corals) and to marine communities. In Australia, the preferred method to establish water quality guidelines for assessing risk involves the development of species sensitivity distributions (SSDs) from individual toxicity thresholds for multiple species representing a community (*Belanger et al., 2016*). The criteria for developing the guidelines have recently been updated in the Australian context and are comprehensively described in *Warne et al. (2018)*. However, many of the alternative pesticides have not been provided Default Guideline Values in the current Australian and New Zealand water quality guidelines (*ANZG, 2018*). To address this, new guideline updates were proposed for 27 GBR-relevant pesticides based on all recent available data (*King et al., 2017a*, *2017b*; *Warne, King & Smith, 2018*); however, many of these values are of low reliability due to lack of toxicity data. Furthermore, the vast majority of data used in the derivation of water quality guidelines have been sourced from tests using temperate and freshwater species (*ANZG, 2018*; *King et al., 2017a*, *2017b*). Clearly, more toxicity data are needed for relevant tropical species to inform the development of water quality guidelines and risk assessments for pesticides that have been detected in sensitive tropical marine habitats.

## Effects of pesticides on corals

Corals represent the key habitat-forming species of tropical coral reefs and can be found adjacent to coastal agriculture in the GBR (*Gilbert & Brodie, 2001*; *Thorburn, Wilkinson & Silburn, 2013*) and globally (*Donner & Potere, 2007*; *Salvat, 1992*). The effects of PSII herbicides on corals primarily impact the photosynthetic capacity of coral symbionts, leading to the breakdown of this symbiosis (bleaching) (*Jones, 2005*; *Jones & Kerswell, 2003*; *Negri et al., 2011a*; *Owen et al., 2002*). This breakdown can have flow-on effects, such as reduced reproductive output (*Cantin, Negri & Willis, 2007*). The effects of insecticides and fungicides on corals are far less studied and are more likely to affect the animal host directly, including the sensitive early life transitions and stages, such as fertilisation, attachment and metamorphosis of planulae larvae into sessile polyps (*van Dam et al., 2011*). *Markey et al. (2007)* examined the effects of insecticides and a fungicide on coral fertilisation and larval metamorphosis. The fungicide MEMC inhibited fertilisation and metamorphosis success at concentrations as low as one $\mu$g L$^{-1}$. While fertilisation was not affected by any of the insecticides (chlorpyrifos, profenofos, endosulfan and permethrin) at concentrations up to 30 $\mu$g L$^{-1}$, metamorphosis was more sensitive to the insecticides, which was inhibited at insecticide concentrations as low as three $\mu$g L$^{-1}$. *Acevedo (1991)* reported mortality in *Pocillopora damicornis* larvae exposed to chlorpyrifos and carbaryl at much higher concentrations (one and 100 mg L$^{-1}$, respectively) but did not test effects on larval function (i.e. attachment and metamorphosis). The organophosphate insecticide naled (Dibrom) reduced the survival of larvae of *Porites astreoides* at three $\mu$g L$^{-1}$ while permethrin had no effect on larval survival or metamorphosis at up to six $\mu$g L$^{-1}$ (*Ross et al., 2015*). The tissues of juvenile *A. tenuis* became partially detached when exposed to the organophosphate dichlorvos at concentrations of 1,000 $\mu$g L$^{-1}$
(*Watanabe, Yuyama & Yasumura, 2006*), a response similar to 'polyp bailout', previously reported as a stress response of stony corals to escape distressing and unfavourable conditions (*Sammarco, 1982*). Only three studies have tested the effects of insecticides on adult corals. The first used a commercial formulation of chlorpyrifos and reported coral mortality at six μg L$^{-1}$ of the active ingredient (*Te, 1998*). *Negri et al. (2009)* found no effects of the biological insecticide *Bacillus thuringiensis* on adult and larval corals and sponges at 5,000 μg L$^{-1}$. A recent study from *Wecker et al. (2018)* reported that the organochlorine insecticide chlordecone caused adult *Pocillopora damicornis* branchlets to release their polyps after 96 h exposure to 30 μg L$^{-1}$ (again similar to 'polyp bailout').

In total, the toxicity of eight insecticides and one fungicide have been assessed on coral larvae but, few of these toxicity data meet the experimental criteria required for inclusion into national water quality guidelines ((*Warne et al., 2018*) e.g. did not use pure compounds, did not measure dissolved pesticide concentrations, etc.). In order to improve water quality guidelines and subsequent risk assessments for pesticides in tropical marine ecosystems, the aim of this current study was to identify the toxicity thresholds of three insecticides (diazinon, fipronil, imidacloprid) and two fungicides (chlorothalonil and propiconazole), which have been detected in the GBR lagoon or catchments, on larval metamorphosis of the common reef-building coral *Acropora tenuis* larvae following 48 h exposures.

## MATERIALS AND METHODS

### Sample collection and larval culture

Gravid colonies (25–40 cm diameter) of the coral *A. tenuis* (Dana, 1846) were collected from 4–8 m depth on two occasions: in November 2016 from Trunk Reef (18°18.2′ S, 146°52.2′ E) and in November 2017 from Falcon Island (18°46′ S, 146°32′ E), GBR under GBRMPA Permit G12/35236.1. Colonies were transported to the National Sea Simulator (SeaSim) at the Australian Institute of Marine Science (AIMS) in Townsville and maintained in 1700 L flow-through holding tanks until spawning. Temperatures were held at 26–27 °C, which was equivalent to the water temperature at the collection sites. Gametes, on both occasions, were collected from eight parental colonies, fertilised and symbiont-free larvae were cultured at approximately 500 larvae L$^{-1}$ in 500 L flow-through tanks (*Negri & Heyward, 2001*; *Nordborg et al., 2018*). Larvae (each 800–1000 μm in length) were competent to undergo attachment and metamorphosis after 5 days and we applied 10-day-old *A. tenuis* larvae on the first occasion and 7-day-old larvae on the second occasion in pesticide exposure experiments. Metamorphosis was defined here as the change in life stage from free swimming or casually attached sausage-shaped larvae (Fig. 1A) to squat, firmly attached, disc-shaped structures with pronounced flattening of the oral–aboral axis and with septal mesenteries radiating from the central mouth region (Fig. 1B) (*Heyward & Negri, 1999*). All pesticides were included in the first experiment; however, minimal inhibition was observed at the highest propiconazole concentration tested. Therefore, a second experiment was run the following year with higher propiconazole concentrations, reference copper and seawater and solvent controls.

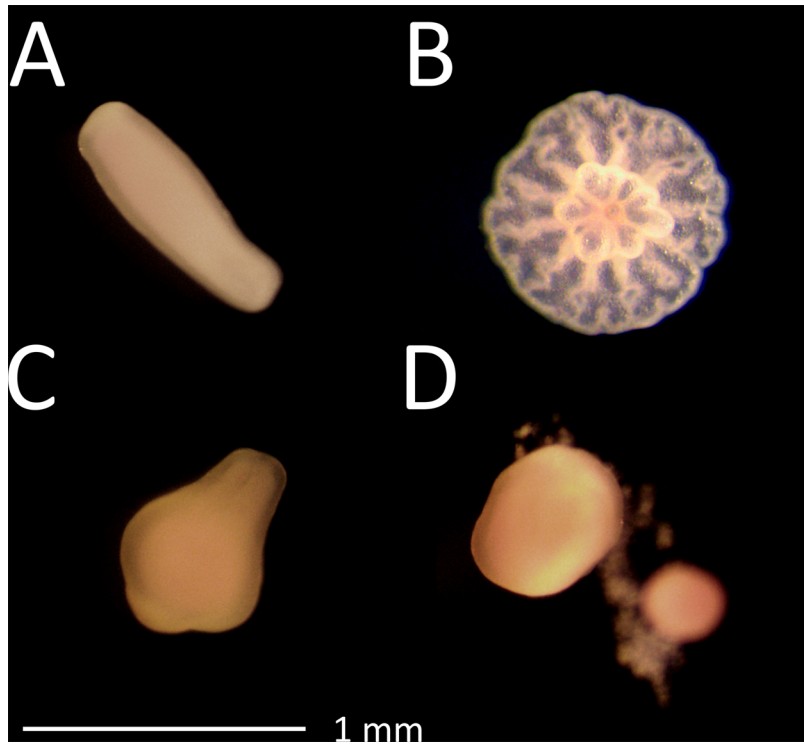

**Figure 1 Photographs after 48 h exposure.** (A) Planula larvae in control treatment; (B) attached post-metamorphosis polyp in control treatment; (C) larvae exposed to 228 µg L$^{-1}$ propiconazole showing slightly abnormal shape but still moving and (D) larvae exposed to 56.3 µg L$^{-1}$ chlorothalonil showing rupturing of cells.

## Pesticides

The five pesticides in this study represent three chemical classes of insecticides and two classes of fungicides (See Table 1). All pesticides were analytical grade (> 98% purity) and purchased from Sigma-Aldrich (Castle Hill, NSW, Australia). Stock solutions (five mg L$^{-1}$) of all pesticides were dissolved in dimethyl sulfoxide (DMSO, 0.01% v/v) and prepared in milli-Q® water. *A. tenuis* larvae were exposed to measured concentrations of diazinon (2.6–638 µg L$^{-1}$), fipronil (1.1–1,144 µg L$^{-1}$), imidacloprid (3.8–947 µg $^{-1}$), propiconazole (7.9–1,975 µg L$^{-1}$) and chlorothalonil (0.5–507 µg L$^{-1}$). A table with the measured concentrations of each pesticide can be found in *eAtlas (2020)*.

## Metamorphosis assays

Static exposures were conducted in 20 mL clear glass scintillation vials containing 12–14 larvae made up to 10 mL filtered seawater (0.5 µm) with six to seven concentrations (per pesticide) and six replicate vials per concentration. All tests included solvent controls containing identical concentrations of DMSO carrier. Seawater and solvent carrier controls were run in 12–18 replicate vials. Copper (CuCl$_2$) was used as a reference toxicant at six measured concentrations between 1.1–35 µg L$^{-1}$ and six replicate vials per concentration.

**Table 1 Pesticides used in present study.**

| Pesticide | Type and class of pesticide | Mode of action | Target pests | Water solubility in mg L$^{-1}$ (log $K_{ow}$ at 25 °C) |
|---|---|---|---|---|
| Diazinon | Insecticide - Organophosphate | Inhibits acetylcholinesterase from breaking down acetylcholine, which leads to continual nerve stimulation (*Cox, 2000*; *Garber & Steeger, 2008*) | Lice, buffalo fly, mange; ectoparasites (*APVMA, 2003*; *ANZECC & ARMCANZ, 2000*) | 40 (3.81) |
| Fipronil | Insecticide - Phenylpyrazole | Blocks the neurotransmitter gamma-aminobutyric acid (GABA) and glutamate-gated chloride channels, causing hyperexcitation of the central nervous system (*Anadon & Gupta, 2012*; *APVMA, 2009*; *Gunasekara et al., 2007*; *Stenersen, 2004*) | Locusts, grasshoppers (*APVMA, 2012*) | 2 (4.0) |
| Imidacloprid | Insecticide - Neonicotinoid | Irreversibly binds to postsynaptic nicotinic acetylcholine receptors disrupting normal neural transmission (*Abbink, 1991*; *Wismer, 2004*; *Stenersen, 2004*) | Canegrub (*Davis et al., 2008*; *Devlin et al., 2015*; *King et al., 2017a*) | 610 (0.57) |
| Chlorothalonil | Fungicide - Organochlorine | Chemically reduces the antioxidant glutathione; enzymes that are dependent on glutathione, including enzymes important in cellular function, become non-functional (*Cox, 1997*; *Raman, 2014*) | Fungal diseases of cereals, fruits and vegetables (e.g. wheat, stone fruit, strawberries, potatoes) and other crops (peanuts, tobacco) (*King et al., 2017b*) | 0.81 (3.05) |
| Propiconazole | Fungicide – Triazole | Inhibits ergosterol biosynthesis critical to the formation of cell walls of fungi, thus inhibiting fungal growth (*USEPA, 2006*) | Rice blast fungus, pineapple sett rot, rust fungi, fungal diseases of bananas, oats, peanuts, perennial ryegrass, stone fruit, sugar cane, wheat (*Bhuiyan, Croft & Tucker, 2014*; *Garland, Davies & Menary, 2004*; *Pak et al., 2017*) | 100 (3.72) |

**Note:**
Pesticides used in this study, their class, mode of action and target pests in Australia. Water solubilities and log $K_{OW}$ values from PubChem Database (National Center for Biotechnology Information, https://pubchem.ncbi.nlm.nih.gov/compound/ (accessed on Apr. 20, 2020)).

Glass vials were transferred in random positions within a refrigerated shaking incubator (TLM-530; Thermoline Scientific, Wetherill Park, NSW, Australia) at 70 RPM to maintain gentle water movement which prevented larvae from attaching and undergoing metamorphosis in the containers (*Negri et al., 2016*). Larvae were exposed under a light intensity of approximately 60 µmol photons m$^{-2}$ s$^{-1}$ (12:12 h light:dark cycle) and at a temperature of 26.7 ± 0.3 °C (mean ± SE). Vials were re-randomised at 24 h. After 48 h exposure larvae and corresponding treatment water were transferred in the same seawater into untreated 6-well polystyrene culture plates (Nunc, Rochester, NY, USA) and returned to the incubator for an additional 24 h but without water movement. Metamorphosis was initiated by the addition of crustose coralline algae (CCA) extract (10 µL) prepared from four g CCA *Porolithon onkodes* (*Heyward & Negri, 1999*; *Negri et al., 2005*). Metamorphosis was assessed after a further 24 h and considered normal if larvae had undergone irreversible attachment to the well plate and had undergone metamorphosis into a polyp form as described above. All other larvae (swimming, casually attached, dead and partially disintegrated) were classed as not metamorphosed. Average

metamorphosis success ≥ 70% in controls was considered indicative of a standard response to metamorphosis inducers based on several previous studies using CCA or extracts of CCA to initiate metamorphosis in coral larvae (*Negri et al., 2016*, *2011b*; *Negri & Heyward, 2000*).

## Chemical analysis and water quality

Analytical samples (two to three samples per pesticide including solvent control) were measured at initiation and termination of experiment. Aliquots (one mL) were transferred into 1.5 mL liquid chromatography amber glass vials and spiked with surrogate standards (i.e. diazinon-d10, 13C4-fipronil, imidacloprid-d4, propiconazole-d5 (stock solution of one µg mL$^{-1}$)). The final concentration of the surrogate standard was 10 ng mL$^{-1}$. The measured concentrations for remaining treatments were calculated based on a linear relationship between nominal and the time weighted average (time = 0 and 48 h) of measured concentrations. All pesticide analyses (except for chlorothalonil) were performed at the Queensland Alliance for Environmental Health Sciences (QAEHS), The University of Queensland using HPLC-MS/MS (SCIEX Triple QuadTM 6500 QTRAP$^{®}$ mass spectrometer Shimadzu Nexera X2 uHPLC system) (*Mercurio, 2016*; *Mercurio et al., 2015*). Queensland Health Forensic and Scientific Services (NATA accreditation No. 41) measured chlorothalonil samples and an external calibration was used for chlorothalonil, so no internal standard was used.

Physico-chemical parameters (pH, salinity and dissolved oxygen) were measured in clear glass containers (60 mL) with approximately 60 larvae in 50 mL filtered seawater using three to six concentrations per pesticide (including controls). Containers were placed alongside experimental vials, and physico-chemical parameters were measured at the start and 48 h post-exposure. Temperature was logged at 5 min intervals (HOBO pendant 64K data logger, Onset Computer Corp, USA). Salinity and pH were measured via a handheld meter (Horiba LAQUAact PC110; Hach, Loveland, CO, USA) and dissolved oxygen concentration was determined with a handheld meter (HQ30d equipped with Intellical LDO101 oxygen probe; Hach, Loveland, CO, USA). Physico-chemical parameters (mean ± SE, $n = 60$) were met throughout the experimental period: temperature (26.7 ± 0.3 °C), dissolved oxygen (> 98% mean saturation, 8.1 ± 0.03 mg L$^{-1}$), salinity (36 ± 0.2 psu) and pH (8.17 ± 0.01). All data can be found in *eAtlas (2020)*.

## Data analysis

No effect concentrations (NECs) and effect concentrations, i.e. concentrations of each pesticide that inhibited 10% and 50% of *A. tenuis* larval metamorphosis relative to controls (EC$_{10}$ and EC$_{50}$, respectively), were calculated from the proportion of metamorphosed larvae as a function of log measured concentration of each pesticide. NECs are the preferred statistical estimate of chronic toxicity for guideline value (GV) derivation (*Fox, 2010*; *Warne et al., 2018*). Bayesian binomial segmented-regression models were applied to the data using the package jagsNEC (*Fisher, Ricardo & Fox, 2019*) in R 3.5.3 (*R Core Team, 2017*), and the model fits were evaluated using trace plots and fitted vs. residual plots. For diazinon, fipronil, chlorothalonil and copper, Bayesian beta
segmented-regression models provided better fits and were applied. Models were run using uninformative priors, with 10,000 Markov chain Monte Carlo iterations after an initial 'burn-in' period of 20,000 iterations for five separate chains. See *Thomas et al. (2020)* for further details. Graphical outputs were generated in R.

## RESULTS

### Larval metamorphosis assays

Metamorphosis success for 7-day-old larvae in solvent control treatments was 72.2 ± 6.1% (mean ± SE) and this was not statistically different from the 75.9 ± 4.8% achieved in seawater controls (student's *t*-test: $p = 0.641$). Ten-day-old larval metamorphosis success was similar: 72.6 ±3.7% in solvent control treatments and 77.5 ± 3.6% in seawater controls (student's *t*-test: $p = 0.366$).

   Larvae from most treatments exhibited normal swimming behaviour after 24 and 48 h (not quantified). However, some coral larvae exposed to 17.4 µg $L^{-1}$ copper and 56.3 µg $L^{-1}$ chlorothalonil (Fig. 1D) were not swimming after 24 and 48 h, respectively. Larvae exposed to 34.9 µg $L^{-1}$ copper and 169 µg $L^{-1}$ chlorothalonil (and above) disintegrated. Larvae exposed to all other contaminants remained intact and were moving. All pesticides, and the reference toxicant copper, inhibited metamorphosis of *A. tenuis* larvae as concentrations increased, enabling the fitting of concentration-response relationships (Fig. 2). Chlorothalonil was the most potent pesticide towards *A. tenuis* metamorphosis with an $EC_{10}$ of 2.8 µg $L^{-1}$ and a NEC of 2.4 µg $L^{-1}$ while propiconazole was the least toxic with an $EC_{10}$ of 330 µg $L^{-1}$ and an NEC of 269 µg $L^{-1}$ (Table 2; Fig. 1C). The relative order of pesticide toxicity according to the calculated $EC_{50}$s was: chlorothalonil > fipronil > diazinon > imidacloprid > propiconazole (Table 2).

## DISCUSSION

All pesticides negatively affected coral larval metamorphosis into a sessile polyp. The success of this early life transition is a critical step in the recruitment process of corals leading to population maintenance and recovery following disturbance (*Harrison & Wallace, 1990*); therefore, inhibition of metamorphosis success is a key endpoint of ecological relevance for assessing risk to coral reef communities. This study is among the first to derive insecticide and fungicide toxicity thresholds for coral that are suitable for national water quality guideline value (WQGV) derivation, revealing differences in toxicity across two orders of magnitude among pesticides (NEC of 2.4 µg $L^{-1}$ for the broad-spectrum fungicide chlorothalonil to a NEC of 263 µg $L^{-1}$ for the insecticide imidacloprid). The specificity of the modes of action of these pesticides provide guidance for assessing their potential to impact non-target species, such as corals. Insecticides and fungicides are designed to affect insects or fungi; however, they can also become toxic to non-target species (such as corals) in several ways including: (i) if coral shares the same receptor/target that the pesticide is designed to affect; (ii) if other specific pathways or cellular processes are inadvertently affected, or (iii) by non-specific narcosis, where hydrophobic contaminants can accumulate in cell membranes and affect structure and function (*Verhaar, van Leeuwen & Hermens, 1992*). The bioassays were applied on 7- and

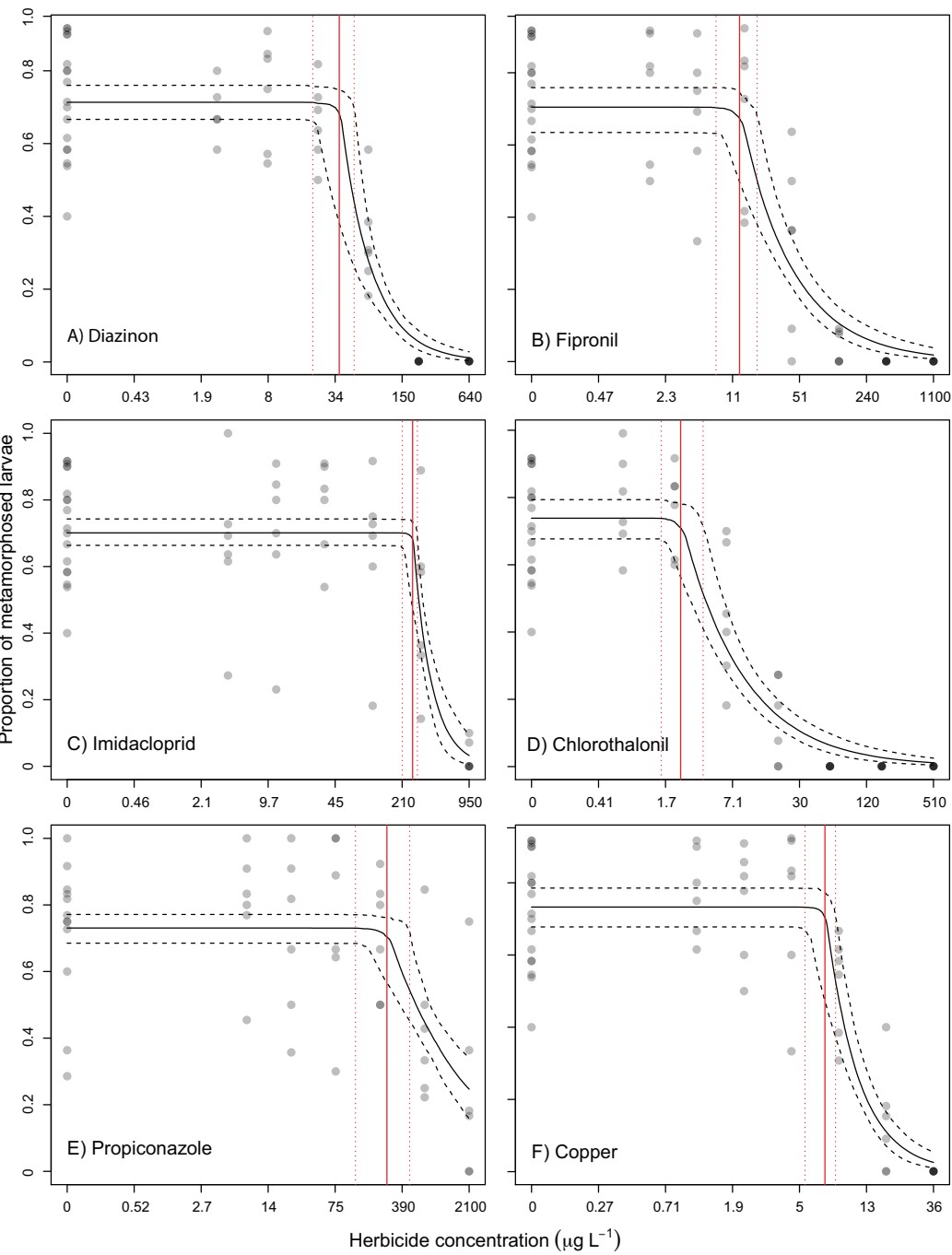

**Figure 2 Concentration-response relationships.** Concentration-response relationships for the toxicity of five pesticides and the reference toxicant copper to *A. tenuis* coral larval metamorphosis. Beta segmented-regression model fits (binomial segmented-regression model fits for imidacloprid and propiconazole) on the proportional decline of coral larval metamorphosis relative to the solvent control treatment (solid black line) and 95% credible intervals (dashed black line) to derive no effect concentrations (red line) and 95% credible intervals (dashed red line) of (A) diazinon; (B) fipronil; (C) imidacloprid; (D) chlorothalonil; (E) propiconazole; (F) copper.

**Table 2 Toxicity estimates.**

| | NEC | $EC_{10}$ | $EC_{50}$ | Meta. (% ± SE) in controls |
|---|---|---|---|---|
| Diazinon | 38.0 (20.4–51.3) | 40.8 (22.4–53.8) | 54.7 (52.3–57.0) | 72.6 ± 3.7 |
| Fipronil | 12.3 (7.1–19.1) | 13.9 (8.5–21.1) | 29.1 (20.2–41.6) | 72.6 ± 3.7 |
| Imidacloprid | 263 (200–295) | 273 (211–306) | 347 (306–417) | 72.6 ± 3.7 |
| Chlorothalonil | 2.42 (1.63–3.89) | 2.76 (1.90–4.42) | 5.95 (4.40–8.82) | 72.6 ± 3.7 |
| Propiconazole | 269 (123 – 468) | 330 (171–537) | 1,008 (704–1689) | 72.2 ± 6.1 |
| Copper | 7.41 (5.75–8.45) | 7.79 (6.13–8.82) | 10.2 (8.6–11.5) | 72.6 ± 3.7 |

Note:
Modelled toxicity estimates for the inhibition of coral larval metamorphosis by diazinon, fipronil, imidacloprid, chlorothalonil, propiconazole and copper to *Acropora tenuis*. No effect concentrations (NECs) and effect concentrations ($EC_{10}$ and $EC_{50}$) were calculated from concentration-response curves (see Fig. 2). Meta. = mean larval metamorphosis (%) of larvae in uncontaminated treatments (± SE). All concentrations are in µg $L^{-1}$ (95% confidence intervals).

10-day-old coral larvae, which at this point in their development were motile (propelled by cilia) and competent to settle (had developed all the cellular structures necessary to detect a chemical cue that initiates metamorphosis into a sessile polyp) (*Babcock & Heyward, 1986*; *Heyward & Negri, 1999*). It is possible that the pesticides which are toxic at low concentrations may directly affect larval biochemistry, including the cellular signalling and processes of metamorphosis. Even if the coral does not have a pesticide-specific target, a hydrophobic pesticide may elicit narcotic toxicity at higher concentrations, with more hydrophobic compounds (measured by the partitioning coefficient between octanol and water, $K_{OW}$) able to accumulate to a greater degree in lipid membranes and cause greater narcotic toxicity (*Di Toro, McGrath & Hansen, 2000*).

The following sections compare the mechanisms of action of the pesticides tested with cnidarian biochemistry and the sensitivity we found in the larval metamorphosis assays. This allowed us to consider the possibility that larvae were affected by the primary modes of action of each pesticide.

## Diazinon

Diazinon affected metamorphosis success of *A. tenuis* coral larvae at relatively low concentrations (NEC: 38.0 µg $L^{-1}$; Table 2). Organophosphate pesticides, like diazinon, inhibit acetylcholinesterase (AChE), an enzyme involved in the breakdown of the chemical acetylcholine (ACh) (*Cox, 2000*; *Garber & Steeger, 2008*). Inhibition of AChE leads to an accumulation of ACh, which results in hyperstimulation and disrupted neurotransmission. Cnidarians, including Hydrozoa and Anthozoa such as corals, possess ACh (*Horiuchi et al., 2003*; *Kass-Simon & Pierobon, 2007*; *Talesa et al., 1992*), representing some of the first indications of a neural net in this primitive phylum. Several studies have shown that ACh is involved in cnidarian neurotransmission; however, whether the chemical acts as a neuromodulator or neurotransmitter is unclear (*Faltine-Gonzalez & Layden, 2019*; *Kass-Simon & Pierobon, 2007*). *Morgan & Snell (2002)* demonstrated the specific induction of esterase gene in the coral *A. cervicornis* when exposed to the AChE inhibitor mosquitocide Dibrom. The toxicity of diazinon to coral larvae adds to evidence that AChE inhibitors, including the organophosphate chlorpyrifos
(*Acevedo, 1991*; *Markey et al., 2007*) and Dibrom (*Ross et al., 2015*), are toxic to corals. Chlorpyrifos has been shown to disrupt swimming behaviour of larval *Pocillopora damicornis* at 100 μg L$^{-1}$ and a 50–100% larval mortality at 1,000 μg L$^{-1}$ (*Acevedo, 1991*) while adult coral *Pocillopora damicornis* were sensitive to effluent water from soil treated with chlorpyrifos with a 96-h LC$_{50}$ of six μg L$^{-1}$ (*Te, 1998*). *Ross et al. (2015)* found that Dibrom as low as 2.96 μg L$^{-1}$ inhibited larval metamorphosis of the coral *Porites astreoides*. In this study, *A. tenuis* larvae exhibited a similar sensitivity to diazinon as the shrimp *Penaeus duorarum* and sea urchin *Paracentrotus lividus* (21 μg L$^{-1}$ LC$_{50}$, 30 μg L$^{-1}$ LOEC, respectively; Table 3 (*Pesando et al., 2003*; *Sunderam et al., 2000*)). Diazinon concentrations in the GBR catchments and coastal waters have been detected up to 0.7 ng L$^{-1}$ (*O'Brien et al., 2014*) which exceeds the 99% species protection guideline value of 0.03 ng L$^{-1}$ in freshwater systems (Table 4). Due to very limited published marine toxicity data for diazinon, there are no reliable GVs for diazinon in marine ecosystems. Therefore, current WQGVs for diazinon in marine waters uses the freshwater guideline protective concentrations, and these would be protective of coral larval metamorphosis.

## Fipronil

Fipronil, belonging to the phenylpyrazole family, is a systemic insecticide which blocks the gamma-aminobutyric acid (GABA)-regulated chloride channel in neurons, particularly in insects (*Anadon & Gupta, 2012*; *Gunasekara et al., 2007*; *Stenersen, 2004*). GABA itself has been found to be involved in cnidarian neurotransmission (*Kass-Simon & Pierobon, 2007*) and several neurotransmitters have been shown to induce larval metamorphosis (*Moeller, Nietzer & Schupp, 2018*). Crustaceans are one of the most highly sensitive non-target groups to fipronil exposure, which is expected as they belong to the same phylum as insects (Arthropoda). For example, fipronil has been shown to affect survival (20% mortality) of adult female grass shrimp *Palaemonetes pugio* at concentrations as low as 0.2 μg L$^{-1}$ (see Table 3) (*Volz et al., 2003*). While *A. tenuis* larvae were not as sensitive to fipronil (NEC$_{10}$: 12.3 μg L$^{-1}$) as grass shrimp, this was the most toxic insecticide of the three insecticides tested to corals in the present study. Interruption of neurotransmission (and motor control processes) by affecting coral larval chloride channels is a plausible mode of action for fipronil that should be investigated further. While fipronil is not usually included in the analysis suite for marine sample monitoring (e.g. GBRMPA MMP), it has been detected at "'very low' concentrations in GBR catchments (*Gallen et al., 2019*). However, due to the low frequency of detections (and detected concentrations) in GBR catchments, it is unlikely at present that fipronil substantially contributes to the overall pesticide toxicity risk in GBR marine waters. The 99% species protection proposed guideline value (PGV) for fipronil in marine waters is 0.0034 μg L$^{-1}$ (Table 4) which is below the limit of reporting level of 0.02 μg L$^{-1}$ by the GBRMPA MMP and would protect coral larval metamorphosis.

## Imidacloprid

Imidacloprid was the least toxic insecticide to *A. tenuis* larvae in the present study. Imidacloprid is a neonicotinoid that irreversibly binds to postsynaptic nicotinic

**Table 3 Summary of single toxicity studies.**

| Phylum | Species | Life stage | Test duration (in days) | Test endpoint(s) | Effects concentration ($\mu g\ L^{-1}$) | Reference |
|---|---|---|---|---|---|---|
| Diazinon | | | | | | |
| Cnidaria | *Acropora tenuis* | Larvae | 2 | Metamorphosis | 38.0 (NEC) | *Present study* |
| Arthropoda | *Mysidopsis bahia* | Not stated | 4 | Mortality | 6 ($LC_{50}$) | *Sunderam et al. (2000)* |
| Arthropoda | *Penaeus duorarum* | Embryo | 4 | Mortality | 21 ($LC_{50}$) | *Sunderam et al. (2000)* |
| Echinodermata | *Paracentrotus lividus* | Gamete | 0.125 | Fertilisation | 30400 (LOEC) | *Pesando et al. (2003)* |
| Echinodermata | *Paracentrotus lividus* | Larvae | 1 | AChE inhibition | 30 (LOEC) | *Pesando et al. (2003)* |
| Fipronil | | | | | | |
| Cnidaria | *Acropora tenuis* | Larvae | 2 | Metamorphosis | 12.3 (NEC) | *Present study* |
| Arthropoda | *Palaemonetes pugio* | Embryo | 4 | Mortality | 32.0 (LOEC) | *Key et al. (2003)* |
| Arthropoda | *Palaemonetes pugio* | Larvae | 4 | Mortality | 0.50 (LOEC) | *Key et al. (2003)* |
| Arthropoda | *Palaemonetes pugio* | Larvae | 4 | Mortality | 0.68 ($LC_{50}$) | *Key et al. (2007)* |
| Arthropoda | *Palaemonetes pugio* | Adult | 4 | Mortality | 0.13 (LOEC) | *Key et al. (2003)* |
| Arthropoda | *Palaemonetes pugio* | Adult | 45 | Survival, weight, length | 0.098 (chronic NOEC) | *Volz et al. (2003)* |
| Arthropoda | *Amphiascus tenuiremis* | Adult | 4 | Mortality | 6.8 ($LC_{50}$) | *Chandler et al. (2004)* |
| Arthropoda | *Amphiascus tenuiremis* | Nauplii | 12–17 | Mature to adult; egg production | 0.22 (LOEC) | *Chandler et al. (2004)* |
| Arthropoda | *Amphiascus tenuiremis* | Adult (female/male) | 4 | Mortality | 6.07/3.86 ($LC_{50}$) | *Bejarano, Chandler & Decho (2005)* |
| Arthropoda | *Americamysis bahia* | < 24 h | 28 | Mortality | 0.0034 (chr. est. NOEC) | *USEPA (2015)* |
| Arthropoda | *Farfantepenaeus aztecus* | Juvenile | 29 | Mortality | 1.3 (96-h $LC_{50}$) | *Al-Badran et al. (2018)* |
| Rotifera | *Brachionus plicatilis* | Adult | 1 | Population growth | 1000 (NOEC) | *Lee et al. (2018)* |
| Imidacloprid | | | | | | |
| Cnidaria | *Acropora tenuis* | Larvae | 2 | Metamorphosis | 263 (NEC) | *Present study* |
| Arthropoda | *Palaemonetes pugio* | Larvae | 4 | Mortality | 308.8 ($LC_{50}$) | *Key et al. (2007)* |
| Arthropoda | *Palaemonetes pugio* | Adult | 4 | Mortality | 563.5 ($LC_{50}$) | *Key et al. (2007)* |
| Arthropoda | *Callinectes sapidus* | Megalopae | 1 | Mortality | 10.04 ($LC_{50}$) | *Osterberg et al. (2012)* |
| Arthropoda | *Callinectes sapidus* | Juvenile | 1 | Mortality | 1112 ($LC_{50}$) | *Osterberg et al. (2012)* |
| Arthropoda | *Americamysis bahia* | Juvenile | 4 | Mortality | 37.7 ($LC_{50}$) | *USEPA (2015)* |
| Arthropoda | *Americamysis bahia* | Not stated | 4 | Mortality | 38 ($EC_{50}$) | *USEPA (2015)* |
| Arthropoda | *Americamysis bahia* | Not stated | 4 | Mortality | 159 ($EC_{50}$) | *USEPA (2015)* |
| Arthropoda | *Mysidopsis bahia* | Adult | 4 | Mortality | 13.3 (NOEC) | *Ward (1990)* |
| Arthropoda | *Artemia* sp. | Adult | 2 | Mortality | 361000 ($LC_{50}$) | *Song, Stark & Brown (1997)* |
| Chordata | *Menidia beryllina* | Larval | 7 | Growth inhibition | 34000 (LOEC) | *Environment Canada (2005)* |
| Chlorothalonil | | | | | | |
| Cnidaria | *Acropora tenuis* | Larvae | 2 | Metamorphosis | 2.42 (NEC) | *Present study* |
| Chlorophyta | *Dunaliella tertiolecta* | Log growth phase | 4 | Population growth | 33 (NOEC) | *DeLorenzo & Serrano (2003)* |

| Phylum | Species | Life stage | Test duration (in days) | Test endpoint(s) | Effects concentration ($\mu$g L$^{-1}$) | Reference |
|---|---|---|---|---|---|---|
| Arthropoda | *Amphiascus tenuiremis* | Adult (female/male) | 4 | Mortality | 53.1/26.7(LC$_{50}$) | *Bejarano, Chandler & Decho (2005)* |
| Chordata | *Ciona intestinalis* | Embryo | 2 | Embryonic development | 12 (EC$_{10}$) | *Bellas (2006)* |
| Mollusca | *Mytilus edulis* | Embryo | 2 | Embryonic development | 4.5 (EC$_{10}$) | *Bellas (2006)* |
| Echinodermata | *Paracentrotus lividus* | Embryo | 2 | Embryonic development | 4.3 (EC$_{10}$) | *Bellas (2006)* |
| Bacillariophyta | *Skeletonema costatum* | Not stated | 14 | Population growth | 5.9 (chronic NOEL) | *USEPA (2015)* |
| Arthropoda | *Americamysis bahia* | Early life stage | 28 | Mortality | 0.83 (chronic NOEL) | *USEPA (2015)* |
| **Propiconazole** | | | | | | |
| Cnidaria | *Acropora tenuis* | Larvae | 2 | Metamorphosis | 269 (NEC) | *Present study* |
| Mollusca | *Crassostrea virginica* | Spat | 4 | Cell density | 170 | *USEPA (2015)* |
| Arthropoda | *Americamysis bahia* | Not stated | 4 | Mortality | 51 | *USEPA (2015)* |
| Chlorophyta | *Dunaliella tertiolecta* | Log growth phase | 4 | Population growth | 375 (NOEC) | *Baird & DeLorenzo (2010)* |
| Bacillariophyta | *Skeletonema costatum* | Not stated | 11 | Population growth | 5.5 (chr. est. NOEC) | *USEPA (2015)* |
| Chordata | *Cyprinodon variegatus* | Early life stage | 100 | Mortality | 150 (NOEL) | *USEPA (2015)* |

**Notes:**

Summary of a selection of single toxicity studies (estuarine and marine) using diazinon, fipronil, imidacloprid, chlorothalonil and propiconazole. Adapted from *King et al. (2017a, 2017b)*.

NEC, no effect concentration; LOEC, lowest observed effect concentration; NOEC, no observed effect concentration; chr. est. NOEC, chronic estimated no observed effect concentration; NOEL, no observable effect level.

**Table 4 WQGV and PGV.**

| Pesticide | WQGV | | | | PGV | | | | Guideline reliability | NEC—this study |
|---|---|---|---|---|---|---|---|---|---|---|
| | PC99 | PC95 | PC90 | PC80 | PC99 | PC95 | PC90 | PC80 | | |
| Diazinon* | 0.00003 | 0.01 | 0.2 | 2.0 | | | | | Unknown | 38.0 |
| Fipronil** | | | | | 0.0034 | 0.0089 | 0.016 | 0.033 | Moderate | 12.3 |
| Imidacloprid*** | | | | | 0.057 | 0.13 | 0.23 | 0.46 | Moderate | 263 |
| Chlorothalonil | | | | | 0.34 | 1.0 | 1.7 | 2.9 | Moderate | 2.42 |
| Propiconazole | | | | | 2.1 | 8.2 | 15 | 30 | Low | 269 |

**Notes:**

Australian water quality guideline values (from *ANZG, 2018*) for diazinon and Department of Environment and Science (DES) proposed guideline values for fipronil, imidacloprid, chlorothalonil and propiconazole for 99%, 95%, 90% and 80% species protection in marine ecosystems (from *King et al., 2017a, 2017b*).

All concentrations are in $\mu$g L$^{-1}$ (95% confidence intervals).

WQGV, water quality guideline value; PGV, proposed guideline value; NEC, no effect concentration.

* Diazinon WQGVs derived from freshwater data only as insufficient marine toxicity data to derive reliable guideline value.

** Fipronil PGVs includes toxicity data from six freshwater species. No toxicity data were found for fipronil to Australian and/or New Zealand marine species.

*** Imidacloprid PGVs were calculated from toxicity data from only arthropods due to bimodality in SSD. Data includes toxicity data from two freshwater species.

acetylcholine receptors (nAChRs), interfering with neural transmission (*Abbink, 1991*; *Wismer, 2004*). More specifically, imidacloprid has a narrow specificity for a unique binding subsite of insect nAChRs (*Casida & Quistad, 2004*; *Tomizawa & Casida, 2005*).

Therefore, even though nAChRs have been well described in cnidarians (*Anctil, 2009*; *Chapman et al., 2010*; *Faltine-Gonzalez & Layden, 2019*), imidacloprid's selectivity for insect nicotinic receptors suggests toxicity of imidacloprid to corals may instead be due to general narcotic toxicity which increases with its ability to accumulate in lipid membranes (*Verhaar, van Leeuwen & Hermens, 1992*). However, imidacloprid has a very low log $K_{ow}$ (0.57) indicating low hydrophobicity and is unlikely to bioaccumulate in the lipid membranes of animals. It is unclear whether narcotic toxicity contributed to the effects on coral larvae at the high concentrations of imidacloprid observed here. While there are limited toxicity data for marine species, the blue crab *Callinectes sapidus* megalopae was more than an order of magnitude more sensitive to imidacloprid with an $LC_{50}$ of 10 µg $L^{-1}$ compared to *A. tenuis* coral larvae with an $EC_{10}$ of 263 µg $L^{-1}$ (see Table 3). In contrast, imidacloprid was found to be virtually non-toxic to *Artemia* sp. with an $LC_{50}$ of 361,000 µg $L^{-1}$ (see Table 3). Several recent studies revealed sublethal effects of low concentrations (five µg $L^{-1}$) of imidacloprid on marine species, including reduced lipid content and weight, in the shrimp *Penaeus monodon* over 21 days exposures (*Butcherine et al., 2020*). Sublethal effects of 10 µg $L^{-1}$ imidacloprid over 14 days exposures also affected enzyme activity and fatty acid composition in the Sydney rock oysters *Saccostrea glomerata* (*Ewere, Reichelt-Brushett & Benkendorff, 2019*). Imidacloprid has been detected at concentrations as high as 1.5 µg $L^{-1}$ in groundwater samples (*Devlin et al., 2015*) and up to 0.09 µg $L^{-1}$ in marine samples (*O'Brien et al., 2015*) which exceed the PC99 guideline value (PGV) of 0.057 µg $L^{-1}$ (Table 4). The PGVs for imidacloprid (Table 4) would be protective for coral larval metamorphosis; however, these PGVs values use a mixture of both marine and freshwater ecotoxicity data so more data are needed to develop higher reliability WQGVs for marine organisms.

## Chlorothalonil

Chlorothalonil was the most toxic of the five pesticides tested in this study to coral larvae with a NEC of 2.42 µg $L^{-1}$. Chlorothalonil is a non-systemic organochlorine that binds and depletes the antioxidant glutathione (*Cox, 1997*; *Raman, 2014*; *Tillman, Siegel & Long, 1973*), which is found in animals, plants, fungi and some bacteria (*Alanazi, Mostafa & Al-Badr, 2015*). Chlorothalonil (266 µg $L^{-1}$) has been shown to reduce haemocyte functionality (by altering morphology of phagocytes) and apoptosis of blood cells of the tunicate *Botryllus schlosseri* (*Cima, Bragain & Ballarin, 2008*). Depletion of antioxidants (e.g. glutathione) can lead to oxidative stress, subsequently causing cell damage and death (*Cima, Bragain & Ballarin, 2008*; *Palmer & Traylor-Knowles, 2018*). Therefore, it is plausible that a reduction in the antioxidant glutathione impeded larval metamorphosis of *A. tenuis* by interfering with its cellular metabolism. We also observed adverse swimming behaviour of the coral larvae at 56.3 µg $L^{-1}$ chlorothalonil and disintegration at ≥ 169 µg $L^{-1}$. A similar negative effect was also recognised by *Markey et al. (2007)*, whereby the fungicide MEMC caused coral larvae to stop swimming at concentrations ≥ 3 µg $L^{-1}$. In addition, the breakdown product of chlorothalonil, 4-hydroxy-2,5,6-trichloroisophthalonitrile (not quantified in the present study), has the potential to be more toxic than the parent compound (*Cox, 1997*). Chlorothalonil is

considered a pesticide of interest due to its application within GBR catchments even though chlorothalonil has been detected at only very low concentrations (*Devlin et al., 2015*; *Holligan et al., 2017*). The PC99 PGV of chlorothalonil at 0.34 µg L$^{-1}$ (Table 4) would be protective of coral larval metamorphosis.

## Propiconazole

Coral larvae were not as sensitive to the fungicide propiconazole, with a NEC of 269 µg L$^{-1}$, a concentration highly unlikely to be detected in tropical marine waters. Propiconazole inhibits ergosterol synthesis (*USEPA, 2006*), which is a sterol found in fungi but absent in animals, suggesting another mechanism by which propiconazole interferes with coral larval metamorphosis at high concentrations. Studies have shown potential adverse effects of conazoles on cytochrome P450-mediated process in non-target species. For example, propiconazole has been found to interfere with the production of estrogen (*Iyer & Makris, 2010*) and embryonic development in the crustacean *Daphnia magna* (*Kast-Hutcheson, Rider & LeBlanc, 2001*). Long term exposure (≥ 20 days) to propiconazole (≥ 50 µg L$^{-1}$) has also been shown to cause oxidative stress in liver, gill and muscle tissues of rainbow trout *Oncorhynchus mykiss* (*Li et al., 2010*). *Li et al. (2010)* also reported an increase in lipid peroxidation in *O. mykiss* after exposure with propiconazole. Coral planula larvae contain large quantities of lipid (*Richmond, 1987*) and propiconazole may interfere with larval metamorphosis by attacking membrane lipids. Narcotic toxicity is a more likely mechanism here than for imidacloprid as propiconazole is relatively hydrophobic (log $K_{OW}$ of 3.72) suggesting that propiconazole may accumulate in cell membranes affecting structure and function (*Di Toro, McGrath & Hansen, 2000*). The toxicity of propiconazole is also relatively low for other aquatic species but can be highly dependent on the duration of exposure. After 48 h exposure to propiconazole, the LC$_{50}$ of *D. magna* was 9000 µg L$^{-1}$ but this dropped to 180 µg L$^{-1}$ after 96 h exposure (*Ochoa-Acuña et al., 2009*). A study by *Betancourt-Lozano et al. (2006)* found a 24-h LC$_{50}$ of 1167 µg L$^{-1}$ for juvenile Pacific white shrimp *Litopenaeus vannamei*. Propiconazole has recently been detected in marine samples by the GBRMPA MMP but at very low concentrations < 0.001 µg L$^{-1}$ (*Gallen et al., 2019*), not exceeding the PGVs (Table 4) which would be protective of coral larval metamorphosis.

## Copper

The inhibition of *A. tenuis* larval metamorphosis success by the reference toxicant Cu (II) after 48 h (EC$_{50}$: 10.2 µg L$^{-1}$) was similar to previous studies using the same species, 32 µg L$^{-1}$ (EC$_{50}$) after 24 h (*Negri & Hoogenboom, 2011*) and 35 µg L$^{-1}$ (EC$_{50}$) after 48 h (*Reichelt-Brushett & Harrison, 2000*), and to other coral species, including *A. aspera* (EC$_{10}$: 5.8 µg L$^{-1}$) and *Platygyra daedala* (EC$_{10}$: 16 µg L$^{-1}$) (*Gissi et al., 2017*), validating the sensitivity and suitability of the test. The NEC for Cu (II) in the present study was 7.4 µg L$^{-1}$, which is between the P90 and P80 WQGVs for copper in marine waters (*ANZG, 2018*), indicating the inhibition of larval metamorphosis in this species is relatively sensitive in comparison with other taxa. Although copper is an essential element to life at background concentrations, it has a wide range of toxic actions at higher

concentrations, including blocking biological functional groups of proteins and enzymes, displacing other essential metal ions and modifying the confirmation of biomolecules (*Eisler, 1998*). The specific impacts of Cu (II) on larval metamorphosis are unknown but may affect larval function, including mobility, as observed for coral *Goniastrea aspera* by *Reichelt-Brushett & Harrison (2004)*. More specifically, Cu (II) may affect signal transduction pathways related to larval metamorphosis into sessile polyps (*Negri & Hoogenboom, 2011*).

## Environmental relevance and conclusions

All five pesticides investigated here are commonly applied in agriculture within catchments of the GBR and were toxic to coral larval metamorphosis. While the toxicity thresholds (NECs and $EC_{10}s$) were higher than measured or expected concentrations in tropical waters, these pesticides have the potential to contribute to overall risk posed by pesticide mixtures which are commonly detected in the environment (*Gallen et al., 2019*; *Lewis et al., 2009*; *Warne, Smith & Turner, 2020*). For instance, *Key et al. (2007)* found greater than additive toxicity to grass shrimp *Palaemonetes pugio* larvae when atrazine was added to a fipronil/imidacloprid mixture. The cumulative risks posed by co-occurring pesticides should be assessed by combining the total risks of all pesticides detected and this can be achieved by applying the multisubstance-Potentially Affected Fraction (ms-PAF method) (*Traas et al., 2002*). The effects of pesticides on tropical species can also increase with other pressures common to tropical ecosystems and the ms-PAF method was recently extended to adjust WQGVs for pesticides to account for heatwave conditions (*Negri et al., 2020*). ms-PAF has been applied in pesticide monitoring and reporting in GBR waters and exceedances of WQGVs in GBR waters were more common when the combined toxicity of multiple co-occurring herbicides was considered (*Gallen et al., 2019*). However, high-reliability WQVGs that are necessary to predict ms-PAFs are not available for all alternative pesticides. For example, diazinon is the only pesticide tested in this study with a current Australian and New Zealand guideline (Table 4) and this was derived from temperate freshwater species toxicity data. The proposed GVs for the remaining four pesticides tested in this study (Table 4) include more recent data but rely heavily on toxicity thresholds of temperate freshwater species. Further studies are also needed on the toxicity of their metabolites and transformation products which are also detected in the marine environment, and can be more toxic than the parent compound (*Mercurio et al., 2018*; *Sinclair & Boxall, 2003*). In addition, understanding the toxicities of commercial pesticide formulations need to be more thoroughly investigated as formulations have been found to increase toxic responses in marine organisms (*Devlin et al., 2015*; *Kroon et al., 2015*). For example, *Stoughton (2006)* found the imidacloprid-formulations were more toxic to the fresh/brackish water midge *Chironomus tentans* and the amphipod *Hyalella azteca* than the technical grade imidacloprid. The present study contributes toxicity thresholds for coral that can improve SSDs by the inclusion of this key reef-building taxa, and this will in turn improve the relevance of WQGVs for tropical marine ecosystems of high ecological value.

## ACKNOWLEDGEMENTS

The authors thank the staff at the AIMS National Sea Simulator for field collection assistance and logistical support. The authors acknowledge the Manbarra, Bandjin and Bindal people as the Traditional Owners where this work took place. We pay our respects to their elders past, present and emerging and we acknowledge their continuing spiritual connection to their land and sea country.

### Funding

This research was supported by the Australian Government's National Environmental Science Program (NESP) Tropical Water Quality Hub Project 3.1.5 Ecotoxicology of pesticides on the Great Barrier Reef for guideline development and risk assessments. The funders had no role in study design, data collection and analysis, decision to publish, or preparation of the manuscript.

### Grant Disclosures

The following grant information was disclosed by the authors:
Australian Government's National Environmental Science Program (NESP) Tropical Water Quality Hub Project 3.1.5 Ecotoxicology of pesticides on the Great Barrier Reef for guideline development and risk assessments.

### Competing Interests

The authors declare that they have no competing interests.

### Author Contributions

- Florita Flores conceived and designed the experiments, performed the experiments, analysed the data, prepared figures and/or tables, authored or reviewed drafts of the paper, and approved the final draft.
- Sarit Kaserzon analysed the data, authored or reviewed drafts of the paper, and approved the final draft.
- Gabriele Elisei analysed the data, authored or reviewed drafts of the paper, and approved the final draft.
- Gerard Ricardo analysed the data, prepared figures and/or tables, authored or reviewed drafts of the paper, and approved the final draft.
- Andrew P Negri conceived and designed the experiments, performed the experiments, prepared figures and/or tables, authored or reviewed drafts of the paper, and approved the final draft.

### Field Study Permissions

The following information was supplied relating to field study approvals (i.e. approving body and any reference numbers):
Material was collected under Great Barrier Reef Marine Park Authority (GBRMPA) permit G12/35236.1.

## Data Availability

Data is available at eAtlas: https://eatlas.org.au/data/uuid/da9fc37d-e74b-477d-8cd5-79178cda968c.

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
