# Peer review of "Toxicity thresholds of three insecticides and two fungicides to larvae of the coral Acropora tenuis"

_PeerJ, doi:10.7717/peerj.9615_

## Round 0.1 · original submission · Minor Revisions

Two expert reviewers have evaluated your manuscript and their comments can be seen below. Both have a favourable opinion of your submission, but they also have made some suggestions to improve the manuscript. Please ensure that you attend each of these observations in a revised version of the manuscript.

·

Basic reporting

Very good English
Good ciations, some suggestions made
professional article.
Self contained

Experimental design

within aims and scope of Peer J
well defined and relevant research question
investigated with rigor
methods mostly well define some clarity requested

Validity of the findings

rational clearly stated
data seem robust
clear conclusions

Additional comments

PEER J 48551: Reviewers comments
This is a well written paper that provides timely and useful data. The experimental process has been completed with rigour and is mostly clearly laid out. The results will be useful in setting future guideline values. The graphs and tables are well presented and clear. After reviewing the manuscript, I recommend it for publication after consideration of some minor changes to help clarify methodology, results, and context within existing literature. Further details are below.

Line 75: suggest change ‘need to be included’ to ‘should be included’
Line 90: Suggest changing:
“However, many of the recently detected pesticides are not included in the current Australian and New Zealand water quality guidelines (ANZG 2018).”

to

“However, many of the recently detected pesticides have not been provided Default Guideline Values (DGVs) in the current Australian and New Zealand water quality guidelines (ANZG 2018).”

Line 150: Along with later references to the age of the larvae (e.g. line 217). It is unclear if separate experiments were run, one using 7 day old larvae and the other using 10 day old larvae of if the larvae that were 7 and 10 days old were combines for one experiment. Can this please be made clear.
Also worth noting is this is the age of the larvae at the start of the experiments but as the 48 processes to larvae do get older. It would be good if this could be considered in the text.
Was just one single test completed for each pesticide can you clarify in the text. See also line 247.

Line 159: This section and in other places (e.g. line 175-178, 236) does not make clear the actual endpoint because the words ‘settlement’ and ‘metamorphosis’ are used interchangeable. I suggest a series of photographs which highlight when settlement success was deemed to be achieved and some photos that show when it wasn’t (e.g. loosely attached). Given that there are a series of steps in the process of metamorphosis to settlement the actual ‘endpoint’ needs to be made clear.
Also how were larvae scored if they were not moving or disintegrated? Dead?
I also suggest using the term ‘metamorphosis to settlement’ rather than ‘settlement and metamorphosis’ (consider lines 107, 112, 116, 236, 254, 296, 390, figure 1 caption)

Line 162: Can you provide a table with the measured concentrations used in each experiment.

Line 171-172: Were larvae still being exposed to the contaminants when they were transferred to polystyrene culture plates? Or is this more like a recovery period. Can you clarify the duration of exposure in the detail here.
Also note polystyrene is known to leach materials in the water, were your containers aged?

Line 197: The replicates were in small 20mL vials which cannot fit common DO and conductivity probes. Can the authors describe how (what samples) the physical chemical parameter were determined…e.g. in pooled replicate samples of each dose? And when… At 0 and at 48 hrs, after the additional 24hrs ?

Line 205: it would be interesting to see where significant difference occurred between controls and dose concentrations. Can these be calculated and presented.
Line 276: and further in this paragraph naled and Dibrom are the same pesticide compound so they should be consistently referred to in the same way.

Line 278 suggest rewording:
“The toxicity of diazinon to coral larvae adds to evidence that AChE inhibitors, including the organophosphate chlorpyrifos (Acevedo 1991; Markey et al. 2007) and naled (Ross et al. 2015), are toxic to corals.”
to
“The toxicity of diazinon to coral larvae adds to evidence that AChE inhibitors, including other organophosphate pesticides chlorpyrifos (Acevedo 1991; Markey et al. 2007) and Dibrom (Ross et al. 2015), are toxic to corals.”

Line 282: Change:
“Ross et al (2015) found that as low as 2.96 μg L-1 naled inhibited larval settlement of the coral Porites astreoides.”
to
“Ross et al (2015) found that concentrations as low as 2.96 μg L-1 Dibrom inhibited larval settlement of the coral Porites astreoides.”

Line 285: Provide references sources in text for the studies.

Line 312: References of use to support text and to consider for this section:
P Butcherine, BP Kelaher, MD Taylor, BJ Barkla… - 2020 - Impact of imidacloprid on the nutritional quality of adult black tiger shrimp (Penaeus monodon) Ecotoxicology and Environmental Safety …,

Butcherine, P. Benkendorff, K., Kelaher, B., Barkla, B.J. (2019) The risk of neonicotinoid exposure to shrimp aquaculture. Chemosphere 217, 329-348.

Ewere, E., Reichelt-Brushett, A., Benkendorff, K. (2019) Imidacloprid and formulated product impacts the fatty acids and enzymatic activities in tissues of Sydney rock oysters, Saccostrea glomerata. Marine Environmental Research. 151: 104765.

Ewere, E.E., Powell, D., Rudd, D., Reichelt-Brushett, A., Mouatt, P., Voelcker, N.H., Benkendorff, K. (2019) Uptake, depuration and sublethal effects of the neonicotinoid, imidacloprid, exposure in Sydney rock oysters Chemosphere 23,0 1-13

Line 301 and 335: Clarity required:
See like there is only a fine subtlety between ‘insecticide’ and ‘pesticide’ in these two statements/claims but it is misleading and creates some confusion. Can the text be reworded to provide better clarity?
“While A. tenuis larvae were not as sensitive to fipronil (NEC10: 12.3 μg L-1) as grass shrimp, this was the most toxic insecticide tested to corals in the present study.”
“Chlorothalonil was the most toxic pesticide tested in this study to coral larvae with a NEC of 2.42 μg L-1.”

Line 366-367: it would be good to say why it is inferred that the logkow value would more likely result in narcotic effects…and define ‘more’ that what….
Line 369: maybe ‘duration of exposure’ rather than ‘length of exposure’

Line 376: Reference to consider here and in previous detail relative to larval motility:
Reichelt-Brushett, A. J. and Harrison, P. L. (2004) Development of a sub-lethal test to determine the effects of copper and lead on scleractinian coral larvae. Archives of Environmental Contamination and Toxicology. 47 (1) 40-55.

Reviewer 2 ·

Basic reporting

Clear, unambiguous, and professional English language is used throughout the manuscript. Introduction is clear, with a great logical flow, and provides the necessary background to show the context, relevance and importance of this study. The literature is well referenced & relevant. The structure of the manuscript conforms to PeerJ standards (but see my comment on Introduction subsections). Figures are relevant, high quality, well labelled & described. The raw data are supplied, but not the associated R code to reproduce the analyses.

Experimental design

This study is original primary research, within Scope of the PeerJ journal. The research question is well defined, relevant & meaningful and it is stated how the research fills an identified knowledge gap. Rigorous investigation were performed to a high technical & ethical standard, and methods are described with sufficient detail & information to replicate.

Validity of the findings

All underlying data have been provided; they are robust, statistically sound, & controlled, but see my comment on the control values for the propiconazole, and my comment regarding the findings on the non-quantified swimming behavior. Speculation is present in the discussion and a greater emphasis but should put to identify it as such. Conclusions are well stated, linked to original research question & limited to supporting results.

Additional comments

Overall comment:

This study investigates the effects of 5 pesticides and copper on the settlement and metamorphosis success in Acropora tenuis. This is a very well written study, globally well-constructed (but see my comments regarding the discussion on the need for a better definition of some terms). The topic is well introduced (the sensitivity of coral early life stages to emerging waterborne pollutants), and the scope of the manuscript is well defined (providing baseline and robust data to improve water quality guidelines). The method and the data look robust and are well presented. More specifically, authors have properly controlled the nominal concentrations of the different pesticides that they have used at the beginning and near-end of the exposure experiment, and state to have use adapted and reproducible methods (but see my comment on the Bayesian or binomial models). This study is timely given the current context of coral reefs, and provide strong data and interesting discussion (but see my comment on their speculative aspect) that are currently missing in the literature. Overall, I recommend the acceptance for the publication of this study, if the following comments (see below) are addressed.




Major comments:

a. I recommend to clearly define settlement and metamorphosis, as these terms are used repeatedly, and sometimes interchangeably throughout the manuscript. Settlement and metamorphosis are most often synchronized and associated, in many species. However, in my opinion (and I have no issue with different definitions as long as they are clearly provided) settlement refers to the organism arrival and installation in its new environment (i.e. ecological process, here from planktonic to benthic life), while metamorphosis refers a developmental process (here of a planula into a polype). I suggest the authors to better define these terms and to check throughout the manuscript if the use of “settlement”, “metamorphose” or “settlement and metamorphosis” is best suited. For example, in the Materials and Methods “Settlement assays” subsections, authors write that they assessed metamorphosis after 24h in 6-well plates. So this brings a confusion to the reader whether settlement or metamorphosis was assessed. And actually, both settlement and metamorphosis were assessed as authors state that changing “from free swimming or casually attached pear-shaped forms to squat, firmly attached, disc-shaped structures with pronounced flattening of the oral–aboral axis and with septal mesenteries radiating from the central mouth region” were the parameters considered. I acknowledge that this is very “etymological”, but I believe this would strengthen the clarity of this manuscript to better define this terms and better define what was precisely examined.

b. I recommend the authors to clarify their use of the terms “new”, “alternative”, “emerging”, “recently-detected” when referring to pesticides. From my understanding of the study, these terms are referring to the same thing throughout the manuscript and have been used interchangeably. If this is the case, I suggest choosing a single one of this term and then only use this term throughout the manuscript. Otherwise, I recommend the authors to specify/clarify this, as I acknowledge that it is fine to use multiple terms as long as their use is justified.

c. I am not convinced that introducing studies that looked at the effects of insecticides on adult corals (Introduction, lines 127-133) adds important information to the readers, but rather dilutes the message and disperses my focus, as authors have already guided their introduction towards early-life stages earlier in the paragraph. I recommend the authors to better justify their use or maybe to remove these few lines.

d. The Introduction is overall very clear, well referenced, and the logical flow is well adapted to the topic and very convincing. I am not familiar with subsection titles in the Introduction, and while they are fine to me, I can also say that their removal would not impair the flow of the Introduction given the great piece of writing that the Introduction is.

e. It is not clear to me whether the larvae that the authors have used were 7-10 days old, or 10-13 days old at the time of metamorphosis (given the 3-day exposure duration, i.e. 2 days in glass vials and 1 day in 6-well PS culture plate). This is really not a big deal to me, but since the authors, in the Results section, mention the metamorphosis success for 7-day-old larvae vs. ten-day-old larvae, I guess this is important to fix. From my understanding, authors want here to say that whether larvae were exposed at 7 or 10 days old, their metamorphosis success after exposure (i.e. 3 days after) was similar. If this is the case, I recommend the authors to edit their results and materials and methods sections to make this clearer.

f. I invite the authors to justify the great emphasis on larvae swimming behavior and disintegration observations, in both the abstract and results. The reason I say this is because 1/ these measures were not quantified as stated in the results section, 2/ there is no Materials and Methods about how these observations were made; and 3 / these observations apparently only concern copper and chlorothalonil, but none of the 4 other pesticides. I am not saying that these information/data are not valuable and I commend the authors for bringing such information that are interesting. However, in my opinion, NEC, EC10 and EC50 of the 6 chemicals are the main results of this study and this should be reflected in the abstract and results sections. I therefore suggest that authors put a greater emphasis at describing the NEC, EC10 and EC50 outputs and a lower emphasis at mentioning the swimming behavior and disintegration outputs, in both the abstract and results sections. I also recommend the authors to bring more information about how the swimming behavior and disintegration observations were made, and why only copper and chlorothalonil were concerned by these observations.

g. The two first paragraphs of the Discussion, before the Diazinon subsection, would benefit from a rewording, as the critical information here but, in my opinion, is a bit dispersed and sometimes repetitive (see lines 246 and 260-261). Specifically, I recommend the authors to gather these two paragraphs in one “introductory” discussion paragraph, in which they 1/ briefly summarize their findings (all the tested pesticides affected coral larval settlement/metamorphosis, but with varying toxicity levels as evidenced by the great range of NEC and EC; 2/ introduce the value/importance of their of their study (importance of testing settlement/metamorphosis sensitivity given the critical role of this life-history transition for recruitment and replenishment of populations and the structure of the ecosystem they inhabit or even drive (in the case of corals), the well-known sensitivity of such early life stages to pollutants, in corals but also in all organisms that undergo metamorphosis (~80% of animal species), and value/novelty of their study in this respect and towards better guidelines); and 3/ introduce that the specificity of the mode of action of each pesticide could explain their results and will be discussed in followings subsections.

h. The discussion on the effects and modes of action of the different pesticide is essentially speculative, which is not an issue to me as it is still valuable and overall the discussion is very interesting. However, in my opinion, the sentence used by authors “The following sections compare the mechanisms of action of the pesticides tested with the possibility of a specific response by the coral larvae” does not sufficiently acknowledge that this discussion is speculative, and this should be greater emphasized. Also, if being speculative, why not also going even further and discuss the potential for these pesticides to disrupt the signaling pathways that are responsible for coral metamorphosis? I acknowledge that this is all very speculative and most is unknown regarding these processes, but I found this study to be a great invitation towards a better understanding of the molecular determinants of coral metamorphosis, and the pollutants that could disrupt it.

i. When looking at panel E) Propiconazole in Figure 1, I observe that the starting value (for herbicide concentration of 0 µg L-1), which corresponds to the solvent control if I understood well, are different from the starting values of all the other panels of this figure. When looking at the raw data, I observe that this is due to the fact that different controls were used for propiconazole. Can the authors explain the reason why different controls were used?

j. In the Materials and Methods, authors state that they used “Bayesian or binomial segmented-regression models” but do not justify why one or the other method was applied, and to which dataset each of this method was applied to. I invite the authors to precise this.






Specific and/or minor comments:

- Adding page numbers would help the review.

- Abstract, line 20: I’m not sure that “alternative pesticides” are well known by non-specialists. I suggest rewording this sentence and the previous one to make it clearer to the readers what “alternative pesticides” are, and that they are the same as the “new pesticides” that you mentioned in the previous sentence.

- Introduction, lines 77-78: Is/Are there reference(s) for this or is it already a conclusion of this study?

- Introduction, line 87: There is probably a missing “,” after “Australia”.

- Introduction, lines 105-109: I suggest a slight rewording the second part of this sentence, as it is not “the effects” that are impacting the the photosynthetic capacity, but the herbicides. I also suggest to separate the last of this long sentence in a single sentence, to make this part about the subsequent “reproductive outputs” clearer.

- Introduction, line 107: Add “this” before “symbiosis”.

- Introduction, line 110: I would remove “potentially”.

- Introduction, line 112: Add “planulae” before “metamorphosis”.

- Materials and Methods, line 206: There is an extra “(“ here. I recommend rewording as follow: “No effect concentrations (NECs) and effect concentrations, i.e. concentrations of each pesticide…”

- Materials and Methods, line 221: Remove “statistical package” as “package” is already mentioned before “jagsNEC”.

- SD vs SE. Data +/- SD are used in the Materials and Methods section, while Data +/- SE are used in the Results section. I recommend the authors to be consistent throughout the entire manuscript regarding this.

- Results, line 228: I suggest to remove “inhibiting metamorphosis by 10%” as this is already what the EC10 means, as defined by the authors in their Materials and Methods. A potential rewording could be “Chlorothalonil was the most potent pesticide towards A. tenuis metamorphosis, with an EC10 of …”.

- Discussion, lines 250-252: I would remove this sentence, which is ‘out-of-topic’ in my opinion and thus breaks the logical flow of this paragraph on the importance of sensitivity of settlement/metamorphosis.

- Discussion, line 267: I suggest the authors to be a bit more specific here, e.g. by rewording this sentence by something similar to: “Organophosphate pesticides, like the diazinon, can inhibit acetylcholinesterase …”. This sentence, and the following one, would also benefit from reference(s), there are plenty on coral reef fishes for example.

- Discussion, lines 293-294: This sentence would benefit from reference(s).

- Discussion, lines 317 and 322: “GBRMPA MMP” has been defined in the Introduction section, but is the MMP mentioned at these lines the MMP from the GBRMPA MMP?

- WQGV is defined in Table 4, but not in the discussion text.

- Discussion, line 321: This sentence would benefit from introducing what K0w refers to.

- Discussion, lines 336-338: This sentence would benefit from reference(s).

- Discussion, line 345: There is probably a missing “concentration” word after “at”.

---

## Round 0.2 · Minor Revisions

I am satisfied with the changes made to the manuscript except for a couple of minor corrections, which should be incorporated:
Abstract
L19: change “an additional suite of ‘alternative’ pesticides” to “a suite of ‘alternative’ pesticides”
L30: “this data” should read “these data” (data is plural; datum is singular)
L31: “in turn” is redundant
L109: “has been” should read “have been”
L111: “data is” should read “data are”
Introduction
L131: “fertilisation and the attachment” should read “fertilisation, attachment”
L142: “metamorphosis up” should read “metamorphosis at up”
Materials and methods
L284: “made up to” should read “in”
Discussion
L540: “there is limited” should read “there are limited”
L554: “data is” should read “data are”

---

## Round 0.3 · accepted · Accept

I am satisfied with the minor modifications to the manuscript.